# Dynamic Changes of Volatile Compounds during the Xinyang Maojian Green Tea Manufacturing at an Industrial Scale

**DOI:** 10.3390/foods11172682

**Published:** 2022-09-02

**Authors:** Peng Yin, Jing-Jing Wang, Ya-Shuai Kong, Yao Zhu, Jun-Wei Zhang, Hao Liu, Xiao Wang, Gui-Yi Guo, Guang-Ming Wang, Zhong-Hua Liu

**Affiliations:** 1Henan Key Laboratory of Tea Plant Comprehensive Utilization in South Henan, Henan Engineering Research Center of Tea Processing and Testing, College of Tea Science, Xinyang Agriculture and Forestry University, Xinyang 464000, China; 2Key Laboratory of Tea Science of Ministry of Education, National Research Center of Engineering and Technology for Utilization of Botanical Functional Ingredients, Co-Innovation Center of Education Ministry for Utilization of Botanical Functional Ingredients, Hunan Agricultural University, Changsha 410128, China; 3Xinyang Yunzhen Tea Co., Ltd., Xinyang 464000, China; 4Xinyang Xianfeng Tea Co., Ltd., Xinyang 464000, China; 5Xinyang Wenxin Tea Co., Ltd., Xinyang 464000, China

**Keywords:** Xinyang Maojian (XYMJ) green tea, volatile compounds, key odorants, manufacturing process

## Abstract

Xinyang Maojian (XYMJ) is one of the premium green teas and originates from Xinyang, which is the northernmost green tea production area in China. The special geographic location, environmental conditions, and manufacturing process contribute to the unique flavor and rich nutrition of XYMJ green tea. Aroma is an important quality indicator in XYMJ green tea. In order to illustrate the aroma of XYMJ green tea, the key odorants in XYMJ green tea and their dynamic changes during the manufacturing processes were analyzed by headspace solid-phase microextraction (HS-SPME) combined with gas chromatography-mass spectrometry (GC-MS). A total of 73 volatile compounds of six different chemical classes were identified in the processed XYMJ green tea samples, and the manufacturing processes resulted in the losses of total volatile compounds. Among the identified volatile compounds, twenty-four aroma-active compounds, such as trans-nerolidol, geranylacetone, nonanal, (+)-δ-cadinene, linalool, (Z)-jasmone, cis-3-hexenyl butyrate, cis-3-hexenyl hexanoate, methyl jasmonate, and β-ocimene, were identified as the key odorants of XYMJ green tea based on odor activity value (OAV). The key odorants are mainly volatile terpenes (VTs) and fatty acid-derived volatiles (FADVs). Except for (+)-δ-cadinene, copaene, cis-β-farnesene, (Z,E)-α-farnesene and phytol acetate, the key odorants significantly decreased after fixing. The principal coordinate analysis (PCoA) and the hierarchical cluster analysis (HCA) analyses suggested that fixing was the most important manufacturing process for the aroma formation of XYMJ green tea. These findings of this study provide meaningful information for the manufacturing and quality control of XYMJ green tea.

## 1. Introduction

Tea is one of the most consumed non-alcoholic beverages worldwide after water. Due to its pleasant taste, attractive aroma, and health-promoting effects [1,2,3], more than two billion people in approximately 170 countries and regions drink tea. According to the manufacturing techniques and sensory qualities, tea can be classified into six different types, including green, white, yellow, oolong, black and dark tea [4]. Green tea is a nonfermented tea and is usually produced from small-leaf tea cultivars (*Camellia sinensis* var. *sinensis*) [5]. The primary manufacturing processes for green tea include spreading, fixing, rolling, and drying, which maintain the color of the infusion and leaves green. Fixing is the feature of green tea manufacturing and is crucial for high-quality green tea, which must follow the rules of fast, even, and high-temperature. The aims of fixing are to deactivate enzymes in tea shoots for maintaining a green color, to promote the development of volatile compounds, and reduce moisture content [4]. In China, green teas are classified as roasted, baked, steamed, and sun-dried green tea based on the fixing and/or drying methods [6].

Tea shoots are rich in volatile and non-volatile compounds that determine the flavor quality of processed tea. Manufacturing contributes to the taste and aroma of green tea [7]. Widely targeted metabolomic analysis revealed that the manufacturing of green tea, especially fixing, had an influence on the evolution of the non-volatile and volatile metabolites. Further analysis indicated that amino acids and lipids played a key role in the formation of volatile compounds [8]. Most non-volatile and volatile metabolites varied notably in different manufacturing processes, and the transformation of metabolites was the dominant trend during green tea processing [6]. During green tea manufacturing, especially in the fixation stage, significant lipidomic variations were observed, which potentially contribute to the color and aroma quality of tea [9]. All lipid degradation products, such as (E,E)-2,4-heptadienal, showed the highest concentrations in the fresh tea leaves but significantly lower concentrations in the final green tea [10]. Except for cis-3-hexenyl hexanoate, the concentrations of most aroma compounds decreased after fixation during the Xinyang Maojian (XYMJ) green tea manufacturing [11].

In China, green tea has the longest production history and an excellent quality. There are many world famous green teas, such as West Lake Longjing tea, XYMJ, Huangshan Maofeng, etc. The XYMJ green tea is one of the top ten most popular teas in China, which originates from Xinyang, Henan province, China [12]. With a production history of more than 2300 years and its excellent quality, the regional public brand value of XYMJ green tea has risen to 7.572 billion Yuan (RMB) in the year of 2022 and ranked third after West Lake Longjing tea and Pu’er tea. The manufacturing techniques of XYMJ green tea are traditional manual processing, the common manual-mechanical processing (the combination of manual and mechanical), and continuous mechanical processing [13]. The output of XYMJ green tea manufactured manually and mechanically accounts for the majority.

A few studies have focused on the volatile compounds of XYMJ green tea [14,15] and the key odorants responsible for the XYMJ green tea have been identified [16,17]. To get a deeper insight into the effect of the single manufacturing process on the XYMJ green tea aroma formation, the key odorants and their dynamic changes during manufacturing processes from tea shoots to drying must be elucidated. Headspace solid-phase microextraction (HS-SPME) combined with gas chromatography-mass spectrometry (GC-MS) has been widely applied to determine volatile compounds in teas [18,19,20]. Thus, the aim of the present study was to investigate the key odorants in XYMJ green tea and their dynamic changes during the manufacturing processes using HS-SPME combined with GC-MS. These findings of our study will provide meaningful information for the manufacturing and aroma quality control of XYMJ green tea.

## 2. Materials and Methods

### 2.1. Experimental Materials

The manufacturing of XYMJ green tea was performed by Xinyang Yunzhen Tea Co., Ltd. in Xinyang, China. Slightly different from the traditional manufacturing processes of XYMJ green tea [13,21], the scattering and shaping processes were newly added during the XYMJ green tea manufacturing. Currently, the whole manufacturing process of XYMJ green tea is shown in Figure 1a. (1) Tea shoots. One leaf and one bud of the tea plant (*Camellia sinensis* cv. Xinyang *Quntizhong*) were randomly plucked by tea farmer on 8 April 2021. The tea plantation locates at Heilong pool, Shihe County, Xinyang, China. (2) Spreading. Tea shoots were spread in the bamboo sieves and kept for 4–6 h at the room temperature (about 20 °C). (3) Fixing. The spread tea leaves were fixed for 120 s in a 6CST-80 type roller-hot air fixation machine (Xinyang Yiding Tea Technology Co., Ltd., Xinyang, China) with rolling temperature of 200 °C. (4) Rolling. The fixed tea leaves were rolled for 40 min in a 6CR-45 type rolling machine (Xinyang Yiding Tea Technology Co., Ltd., Xinyang, China). (5) Scattering. The rolled tea leaves were scattered for 15 min at 86 °C in a 6CCGK-7D type scattering machine (Xinyang Yiding Tea Technology Co., Ltd., Xinyang, China). (6) Shaping. The scattered tea leaves were shaped for 6 min at 115 °C in a 6CL-80/16D type shaping machine (Quzhou Hualin Machinery Co., Ltd., Quzhou, China). (7) Drying. The shaped tea leaves were dried for 20 min at 80 °C in a CS-6CHZ-9 type baking machine (Quanzhou Changsheng Tea Machinery Co., Ltd., Quanzhou, China) until the moisture content of tea was approximately up to 6.0%. All the samples were collected after each manufacturing stage, then frozen by liquid nitrogen immediately, vacuum freeze-dried, and packed into the aluminum foil bags and stored at −40 °C for further analysis.

### 2.2. Sensory Evaluations

The sensory evaluation of the XYMJ green tea samples was performed by five experienced tea experts (three women and two men; aged 28–35 years; nonsmokers) [18], and the aroma description and quality scores of the XYMJ green tea samples were assessed according to the national standards Methodology of sensory evaluation of tea (GB/T 23376-2018) [22] and Tea vocabulary for sensory evaluation (GB/T 14487-2017) [23]. To meet the requirements of this study, we only focused on the aroma quality. Accurately, 3.0 g of the tea sample was infused with 150 mL freshly boiled distilled water (100 °C), and filtered after brewing for 4.0 min. The brewed tea leaves were sniffed thrice to evaluate the intensity and persistence of their aroma. The aroma qualities of the XYMJ green tea samples were evaluated using a 100-point scoring scale from 0 to 100 and were determined by calculating the averages of the scores from the five panelists. Each tea sample was assessed three times through blind evaluation.

### 2.3. Extraction of XYMJ Green Tea Volatiles by HS-SPME

All the processed XYMJ green tea samples were ground into a powder in liquid nitrogen, and 0.5 g of the powder was transferred to a 20 mL headspace vial (Agilent, Santa Clara, CA, USA) containing 2.0 mL NaCl saturated solution to inhibit enzyme reactions. Then, 10 μL internal standard solution (3-hexanone-2,2,4,4-D4, 50 μg/mL in anhydrous ethanol) was added immediately. At the time of SPME analysis, each vial was shaken at 100 °C for 5 min, and then a 120 µm DVB/CAR/PDMS microextraction fiber (Supelco, Bellefonte, PA, USA) was exposed to the headspace of the sample for 15 min at 100 °C. The fiber was preconditioned for 5 min in the injection port of the gas chromatograph at 250 °C before analysis. Desorption of the volatile compounds from the fiber coating was carried out in the injection port of the GC apparatus at 250 °C for 5 min in splitless mode.

### 2.4. GC-MS Analysis

The identification and quantification of volatiles were performed by MetWare (http://www.metware.cn/) (accessed on 14 December 2021) using the 8890 GC and 5977B mass spectrometer (Agilent, Santa Clara, CA, USA) equipped with a 30 m × 0.25 mm × 0.25 μm DB-5 MS (5% phenylpolymethylsiloxane) capillary column. Helium was used as the carrier gas at a linear velocity of 1.2 mL/min. The oven temperature was programmed from 40 °C and held for 3.5 min, firstly increased at 10 °C/min to 100 °C, then increased at 7 °C/min to 180 °C, and finally increased at 25 °C/min to 280 °C and held for 5 min. Mass spectra were recorded in electron impact (EI) ionization mode at 70 eV. The quadrupole mass detector, ion source and transfer line temperatures were set at 150, 230, and 280 °C, respectively. Mass spectra were scanned in the range *m*/*z* 50–500 amu at 1 s intervals. Volatile compounds were identified by comparing the mass spectra with the data system library (MWGC) and linear retention index [24,25]. The concentrations of the volatiles were calculated in µg/L based on the internal standard solution.

### 2.5. Odor Activity Value (OAV) Calculation

Odor activity value (OAV) is often applied to evaluate the contributions of aroma compounds. OAV was calculated using the equation OAV = C/OT, where C was the concentration of the volatile compound and OT was its odor threshold in water obtained from references.

### 2.6. Statistical Analysis

All the results were carried out in triplicate for the analytical determination. The analysis of significant differences between the samples was determined by one-way ANOVA (Duncan’s multiple range tests) using SPSS 20.0 (SPSS Inc., Chicago, IL, USA). The principal coordinate analysis (PCoA) and the hierarchical cluster analysis (HCA) were based on Bray–Curtis dissimilarities using vegan package in R (version 4.1, https://www.r-project.org/) (accessed on 16 May 2022).

## 3. Results and Discussion

### 3.1. Sensory Evaluation Analysis

As shown in Figure 1b, the processed XYMJ green tea samples presented various aroma characteristics, including green and floral aroma (tea shoots and spreading), clean aroma (fixing, rolling, scattering, and shaping), and long-lasting chestnut aroma (drying). The chestnut-like aroma is the typical aroma characteristic of some Chinese green tea and is referred to as an important indicator of an excellent-quality green tea [26,27]. Seventeen volatiles were identified as the key odorants responsible for the chestnut-like aroma of green tea, including trans-nerolidol, linalool, nonanal, cis-3-hexenyl hexanoate, 3-methylbutanal, (E)-3-penten-2-one, ethylbenzene and so on [19].

### 3.2. Identification and Quantification of the Volatile Compounds in XYMJ Green Tea

The volatile compounds in each manufacturing process of XYMJ green tea were tentatively identified by HS-SPME/GC-MS. A total of 73 volatile compounds were identified as common compounds for all the tea samples during the XYMJ green tea manufacturing, which were listed in Table 1. According to their chemical structures, these compounds were divided into six different chemical classes, including alcohols, esters, terpenes, aldehydes, ketones, and hydrocarbons. As presented in Figure 2, alcohols were present in the highest concentration (8797.9 μg/L–56,586.2 μg/L), followed by terpenes (11,540.3 μg/L–56,835.6 μg/L), esters (9312.5 μg/L–25,332.2 μg/L), and aldehydes (2835.3 μg/L–24,390.7 μg/L), indicating that they were the four major volatile compound groups. Compared to the other chemical classes, the ketones (1263.8 μg/L–3794.2 μg/L) had the lowest concentration. The concentrations of total volatile compounds in tea shoots and drying stage were 168,218.8 μg/L and 44,851.9 μg/L, respectively, which decreased by 73.3%, indicating that the manufacturing processes resulted in the losses of volatile compounds.

#### 3.2.1. Alcohols

The alcoholic compounds varied during the XYMJ green tea manufacturing, and the changes in the presentative alcoholic compounds were shown in Figure 3. Spreading significantly increased the concentrations of trans-furan linalool oxide, trans-nerolidol, phytol, isophytol, dehydroisophytol, and cis-cubenol. The highest concentrations of trans-furan linalool oxide and trans-nerolidol were 6638.6 μg/L and 4422.4 μg/L at the spreading stage, respectively. Spreading is an indispensable process in the aroma formation of premium green tea. Volatile metabolomics and transcriptomics revealed that the concentration of trans-furan linalool oxide significantly increased after spreading [28]. Except for phytol, isophytol and dehydroisophytol, the fixing significantly decreased the concentrations of other alcoholic compounds. The lowest concentration of linalool was 4265.1 μg/L at the fixing stage. After fixing, remarkable losses were observed for linalool, trans-furan linalool oxide, and trans-nerolidol during green tea manufacturing [10,11,20]. From fixing to drying, linalool and trans-furan linalool oxide, trans-nerolidol, and cis-cubenol, presented the same trends, respectively. The concentrations of phytol, isophytol, and dehydroisophytol gradually increased from spreading to shaping and then decreased at drying. The aroma profiles of green tea made with fresh tea leaves plucked in summer were studied by the analysis of volatile compounds during the manufacturing, which showed that the concentration of phytol in tea shoots was significantly higher than the summer green tea [29], which is inconsistent to the present study. We speculate that the reason might be due to the different manufacturing techniques and the harvested season of tea shoots.

#### 3.2.2. Esters

A total of seventeen esters were identified (Table 1), among which were six hexenyl esters, including cis-3-hexenyl hexanoate, (Z)-3-hexenyl octanoate, cis-3-hexenyl butyrate, cis-3-hexenyl valerate, (E)-2-hexenyl hexanoate, and cis-3-hexenyl crotonate. From tea shoots to drying, cis-3-hexenyl butyrate gradually decreased and reached its minimum level (154.4 μg/L) at drying (Figure 4a), which decreased by 88.5%. cis-3-Hexenyl hexanoate (Figure 4b) and cis-3-hexenyl valerate (Figure 4c) presented a similar trend during the manufacturing, and reached their minimum levels at drying, which decreased by 56.1% and 76.9%, respectively. The concentrations of the six hexenyl esters were the lowest at drying, and the hexenyl esters losses might be due to their high volatility. As shown in Figure 4d,f, the variations of methyl jasmonate and dihydroactinolide appeared a similar trend, and spreading significantly increased their concentrations. Integrated volatile metabolomics and transcriptomics confirmed that spreading could increase the concentration of methyl jasmonate [28]. The high-temperature fixing significantly resulted in the losses of methyl jasmonate and dihydroactinolide, which decreased by 45.5% and 50.5%, respectively. After scattering, the high temperature (115 °C) shaping promoted the increase in the concentrations of methyl jasmonate and dihydroactinolide. Being similar to the above esters mentioned, the concentration of methyl salicylate significantly decreased from tea shoots (11,159.7 μg/L) to fixing (712.9 μg/L). Although the methyl salicylate fluctuated from fixing to drying, there were no significant differences between them. The changing trend of methyl salicylate in this study was consistent with the previous findings [20,30].

#### 3.2.3. Aldehydes and Ketones

The concentration of nonanal was the highest (20,634.3 μg/L) in tea shoots and the lowest (2701.0 μg/L) at fixing, which decreased by 86.9%. From fixing to drying, the concentration of nonanal increased by 18.1% due to the thermal degradation of lipids. As shown in Figure 5a, the dynamic change of nonanal during the XYMJ green tea manufacturing was consistent with the previous studies [11,20]. The concentration of geranial increased to the highest level (4372.1 μg/L) at spreading and sharply decreased to the lowest level (134.3 μg/L) at fixing, which decreased by 96.9%. The concentration of geranial increased from fixing to shaping and then decreased at drying (Figure 5b). The high temperature (200 °C) of fixing resulted in the losses of nonanal and geranial. The variations of nonanal and geranial were no significant differences from fixing to drying.

As presented in Figure 5c, the concentration of (Z)-jasmone significantly increased to 3330.3 μg/L at spreading and then remarkably decreased to 1004.2 μg/L at fixing. It has been reported that spreading contributes to the (Z)-jasmone accumulation, regardless of the tea plant cultivars [28]. Compared to tea shoots (1806.1 μg/L), the concentration of (Z)-jasmone decreased by 51.0% at the drying. The variations of (Z)-jasmone were no significant differences among the subsequent manufacturing processes (from fixing to drying). The concentration of geranylacetone fluctuated during the manufacturing (Figure 5d), which was the highest at spreading (251.8 μg/L) and lowest at rolling (159.2 μg/L). The dynamic change of geranylacetone in the present study was consistent with the previous research [29].

#### 3.2.4. Terpenes

A total of nineteen terpenes were identified (Table 1), and the variations of the representative terpenes were presented in Figure 6. Spreading significantly contributed to the increases of β-ocimene (Figure 6a), β-myrcene (Figure 6b), and D-limonene (Figure 6d), which has been revealed by volatile metabolomics and transcriptomics [28]. From tea shoots to drying, the changing trends of β-ocimene, (E)-β-ocimene, β-myrcene, and D-limonene were similar, which decreased by 90.9%, 91.8%, 92.2%, and 85.6%, respectively. The thermal processes, especially fixing, caused a significant decrease in the four terpenes. From tea shoots to fixing, the fixing significantly increased the concentrations of (Z,E)-α-farnesene (Figure 6c), (+)-δ-cadinene (Figure 6e), and copaene (Figure 6f), which increased by 37.0%, 53.2%, and 252.0%, respectively. For cis-β-farnesene, there were no significant differences (*p* < 0.05) between tea shoots and fixing. From tea shoots to drying, the variations of cis-β-farnesene, (Z,E)-α-farnesene, (+)-δ-cadinene, and copaene presented similar changing trends, which increased by 101.5%, 68.0%, 96.5%, and 356.1%, respectively.

### 3.3. Principal Component Analysis and Hierarchical Cluster Analysis

The identified 73 volatile compounds were used as variables to perform the principal coordinate analysis (PCoA, Figure 7a) and hierarchical clustering analysis (HCA, Figure 7b). The clustering results of PCoA were evaluated by PERMANOVA in vegan, which showed that the processed XYMJ green tea samples were divided into two clusters (PERMANOVA *R*^2^ = 0.976, *p* < 0.001). Cluster-1 consisted of the tea shoots and spreading samples, and Cluster-2 consisted of the remained XYMJ green tea samples from fixing to drying. The results indicated that fixing was the most important manufacturing process for the aroma formation of XYMJ green tea. A total of nine volatile compounds with a variance importance value (VIP) > 1 might contribute to separating the processed XYMJ green tea samples. Linalool had the highest VIP (4.50), followed by β-myrcene (3.75), nonanal (2.99), methyl salicylate (2.39), β-ocimene (2.28), (E)-β-ocimene (1.86), trans-furan linalool oxide (1.75), geranial (1.42) and D-limonene (1.06).

### 3.4. Odor Activity Values

Generally, volatile compounds with OAV ≥ 1 are regarded as potential contributors to an aroma profile [31]. Higher OAVs correspond to a greater contribution to the aroma. Twenty-four volatile compounds had OAVs > 1 in all the processed XYMJ green tea samples, except for phytol acetate with OAV (0.9) in tea shoots (Table 2), and most of the key odorants had floral attributes. The top ten key odorants with the highest OAVs in the made XYMJ green tea were trans-nerolidol (10,199.4), geranylacetone (3436.5), nonanal (3190.1), (+)-δ-cadinene (1881.0), linalool (838.0), (Z)-jasmone (465.6), cis-3-hexenyl butyrate (220.6), cis-3-hexenyl hexanoate (189.0), methyl jasmonate (125.1), and β-ocimene (92.4).

Previous studies and the present result revealed that (E)-nerolidol was the key odorant contributing to the aroma profile of XYMJ green tea and other green teas [11,17,19,32]. Multiple stresses, such as mechanical damage and low temperature, had a synergistic effect on (E)-nerolidol formation during oolong tea manufacturing [33]. (E)-Nerolidol could also be derived from the non-enzymatic degradation of phytofluene [7]. Derived from the carotenoids, the floral geranylacetone was also considered the aroma-active compound in Japanese green tea (Sen-cha) [34], summer green tea [29], and Longjing tea [35]. Based on OAV calculation, nonanal, linalool, (Z)-jasmone, and methyl jasmonate were the key odorants in premium green teas (Longjing, XYMJ, Taiping Houkui, Lu’an Guapian, etc.) [11,35,36,37]. Cis-3-hexenyl hexanoate in XYMJ green tea was determined by both OAV calculation and GC-O, while (+)-δ-cadinene was only detected by OAV calculation, which might be due to the different sample preparation techniques [11]. Therefore, a comprehensive analysis of XYMJ green tea aroma by GC-O technique is required in future studies.

**Table 2 foods-11-02682-t002:** Volatile compounds with OAVs > 1 in the processed XYMJ green tea samples.

Compounds *	Threshold (μg/L) #	Odor Type #	Tea Shoots	Spreading	Fixing	Rolling	Scattering	Shaping	Drying
trans-Nerolidol ^c^	0.25	Floral	15,070.2	17,689.4	10,692.2	7875.2	9909.2	11,838.2	10,199.4
Geranylacetone ^d^	0.06	Floral	4056.3	4197.5	2781.8	2652.8	3355.0	3950.2	3436.5
Nonanal ^a^	1.0	Floral	20,634.3	18,717.2	2701.0	3132.8	3065.1	3292.8	3190.1
(+)-δ-Cadinene ^c^	1.5	Woody	957.2	990.6	1466.0	1397.1	1777.6	1917.3	1881.0
Linalool ^c^	6.0	Floral	7564.9	6605.6	710.8	863.1	813.4	908.8	838.0
(Z)-Jasmone ^a^	1.9	Floral	950.6	1752.8	528.5	494.1	458.7	523.2	465.6
cis-3-Hexenyl butyrate ^a^	0.7	Fruity	1917.0	1136.6	666.8	459.8	273.9	255.9	220.6
cis-3-Hexenyl hexanoate ^a^	16	Green	431.0	298.7	333.0	223.1	217.4	219.9	189.0
Methyl jasmonate ^a^	3.0	Floral	224.5	290.3	122.4	131.3	152.3	230.2	125.1
β-Ocimene ^c^	10	Floral	1014.5	1066.9	57.3	76.0	82.7	96.5	92.4
β-Myrcene ^c^	42	Woody	636.1	696.8	28.8	44.7	45.0	53.2	49.9
Copaene ^c^	7.0	Floral	9.5	7.0	33.3	27.0	37.9	40.4	43.2
cis-β-Farnesene ^c^	87	Floral	17.4	12.0	20.9	15.8	28.8	31.7	35.0
D-Limonene ^c^	10	Fruity	234.2	238.9	20.6	26.1	32.4	33.6	33.8
(E)-β-Ocimene ^c^	18.7	Floral	360.7	374.0	16.0	24.4	25.6	30.3	29.5
Dihydroactinolide ^d^	2.1	Musk	30.5	35.9	15.1	14.9	24.8	31.7	23.0
M-Cymene ^c^	11.4	Floral	68.5	69.1	10.6	10.9	13.4	12.9	13.2
(Z,E)-α-Farnesene ^c^	87	Floral	7.7	6.3	10.6	7.8	11.7	13.2	13.0
Methyl salicylate ^b^	40	Green	279.0	266.9	8.8	12.6	7.0	8.5	6.8
Geranial ^c^	53	Fruity	70.9	82.5	2.5	4.7	5.3	7.4	5.5
Phytol acetate ^e^	750	Floral	0.9	1.8	2.0	2.8	4.0	5.7	4.4
trans-Furan linalool oxide ^c^	190	Floral	31.5	34.9	3.0	3.5	2.9	3.6	3.1
cis-3-Hexenyl valerate ^a^	60	Fruity	11.0	4.2	4.5	3.0	3.0	3.0	2.6
(-)-α-Terpineol ^c^	330	Floral	2.6	2.6	1.2	1.1	1.4	1.7	1.7

* ^a^: fatty acid derived volatiles (FADVs), ^b^: amino acid derived volatiles (AADVs), ^c^: volatile terpenes (VTs), ^d^: carotenoid derived volatiles (CDVs), and ^e^: others. # The threshold values and the odor type were based on the previous literature [11,19,29,36,38,39,40].

As shown in Table 2, the key odorants in the processed XYMJ green tea samples almost belong to endogenous biosynthesis volatiles, including fatty acid-derived volatiles (FADVs), amino acid-derived volatiles (AADVs), volatile terpenes (VTs) and carotenoid-derived volatiles (CDVs) [2,41]. There are fourteen key odorants that are VTs and they present fruity and floral odors, which may contribute to the formation of the superior aroma quality of XYMJ green tea. In addition to cis-3-hexenyl hexanoate with a green odor, the FADVs, including nonanal, (Z)-jasmone, cis-3-hexenyl butyrate, methyl jasmonate, and cis-3-hexenyl valerate, could form the floral quality of XYMJ green tea. As the only AADVs with a green odor, the decrease of the methyl salicylate is beneficial to the formation of the attractive aroma quality of XYMJ green tea. Derived from the carotenoids, geranylacetone and dihydroactinolide are necessary for the formation of the aroma of XYMJ green tea. To sum up, the aroma of XYMJ green tea is the comprehensive presentation of the key odorants, which are derived from multiple biosynthetic pathways.

It has been reported that nine aroma-active compounds mimic the aroma quality of XYMJ green tea, including linalool, geraniol, (E)-nerolidol, nonanal, octanal, decanal, β-cyclocitral, β-ionone, and cis-3-hexenyl hexanoate [11]. Fourteen volatile compounds with high OAVs can recreate the aroma of Longjing green tea [35], and twenty-nine odorants showing OAVs ≥ 1 can simulate the aroma of a high-grade Chinese green tea (*Jingshan cha*) [10]. The twenty volatile compounds with the highest OAVs were used to conduct the aroma shape of Taiping Houkui green tea, and the results suggested that these volatiles might be the key odorants in Taiping Houkui green tea [36]. In the present study, the top twenty odorants with OAVs ≥ 1 in the final XYMJ green tea were used to construct the aroma shape of each processed sample during the XYMJ green tea manufacturing (Figure 8). The result showed that the seven processed samples could be divided into two groups consistent with PCoA and HCA analyses, suggesting that fixing was the crucial process step for the aroma formation of XYMJ green tea.

## 4. Conclusions

In this work, the dynamic changes in the concentration of volatile compounds during the XYMJ green tea manufacturing were analyzed by HS-SPME combined with GC-MS, and the key odorants responsible for the aroma of XYMJ green tea were identified. A total of 73 volatile compounds in six different chemical classes were determined in the processed XYMJ green tea samples. The manufacturing processes resulted in the losses of total volatile compounds, which decreased by 73.3%. Among the identified volatile compounds, twenty-four aroma-active compounds, such as trans-nerolidol (10,199.4), geranylacetone (3436.5), nonanal (3190.1), (+)-δ-cadinene (1881.0), linalool (838.0), (Z)-jasmone (465.6), cis-3-hexenyl butyrate (220.6), cis-3-hexenyl hexanoate (189.0), methyl jasmonate (125.1), and β-ocimene (92.4), were identified as the key odorants of XYMJ green tea based on OAV. The key odorants are mainly VTs and FADVs. Spreading significantly increased the concentrations of trans-nerolidol, trans-furan linalool oxide, methyl jasmonate, dihydroactinolide, (Z)-jasmone, geranial, phytol acetate, β-ocimene, β-myrcene and D-limonene, while significantly decreased the concentrations of nonanal, linalool, cis-3-hexenyl butyrate, cis-3-hexenyl hexanoate, cis-3-hexenyl valerate and cis-β-farnesene. Proper spreading is beneficial for the aroma formation of XYMJ green tea. Except for (+)-δ-cadinene, copaene, cis-β-farnesene, (Z,E)-α-farnesene and phytol acetate, the key odorants significantly decreased after fixing. The PCoA and HCA analyses suggested that fixing was the crucial process step for the aroma formation of XYMJ green tea. These findings of this study provide meaningful information for the manufacturing and quality control of XYMJ green tea.

## Figures and Tables

**Figure 1 foods-11-02682-f001:**
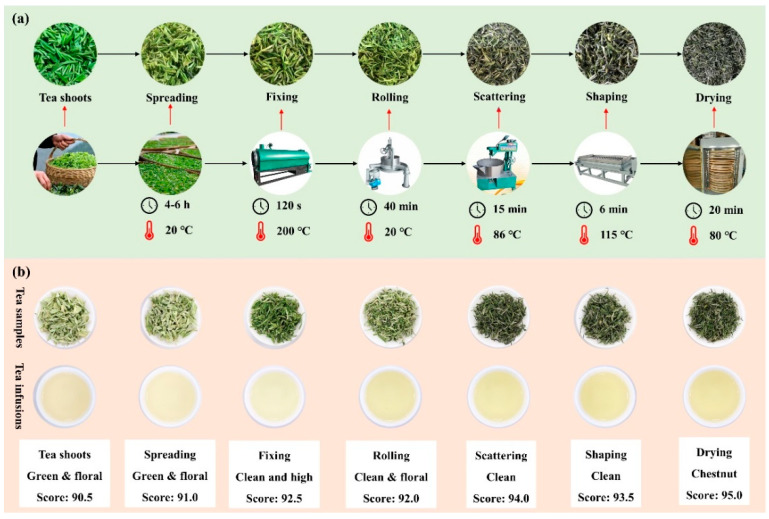
The XYMJ green tea manufacturing flow chart (**a**) and aroma sensory evaluation of the processed XYMJ green tea samples (**b**).

**Figure 2 foods-11-02682-f002:**
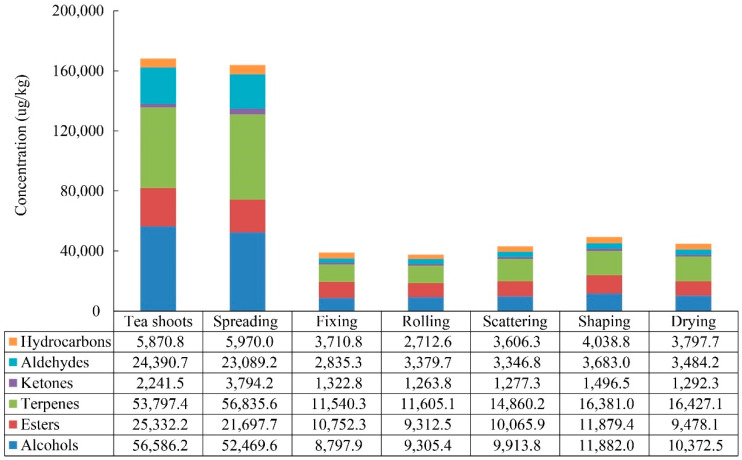
Category and concentrations of volatile compounds in XYMJ green tea during manufacturing.

**Figure 3 foods-11-02682-f003:**
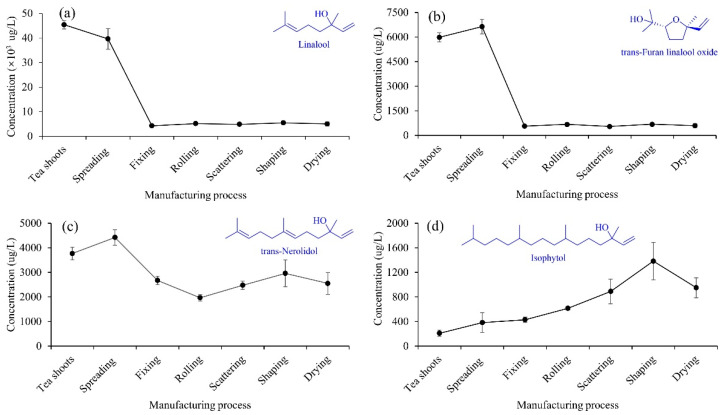
Changes in the concentrations of linalool (**a**), trans-furan linalool oxide (**b**), trans-nerolidol (**c**) and isophytol (**d**) during the XYMJ green tea manufacturing.

**Figure 4 foods-11-02682-f004:**
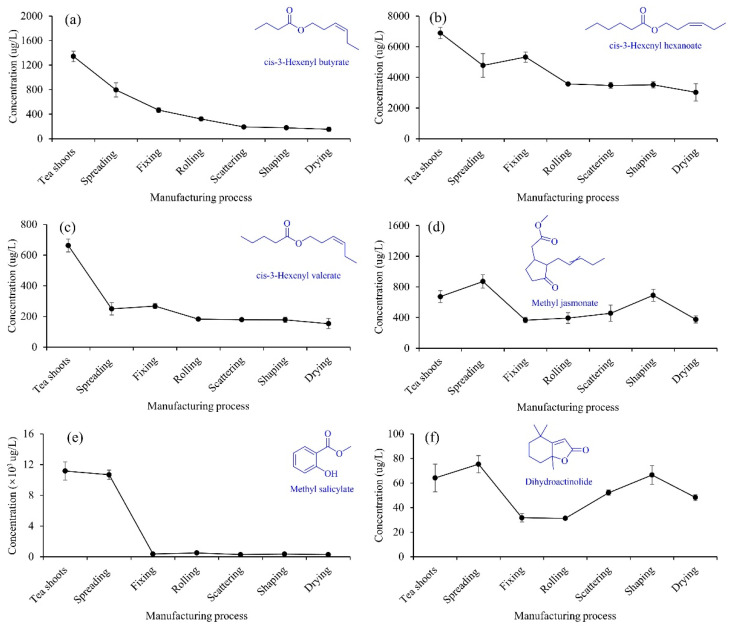
Changes in the concentrations of cis-3-hexenyl butyrate (**a**), cis-3-hexenyl hexanoate (**b**), cis-3-hexenyl valerate (**c**), methyl jasmonate (**d**), methyl salicylate (**e**) and dihydroactinolide (**f**) during the XYMJ green tea manufacturing.

**Figure 5 foods-11-02682-f005:**
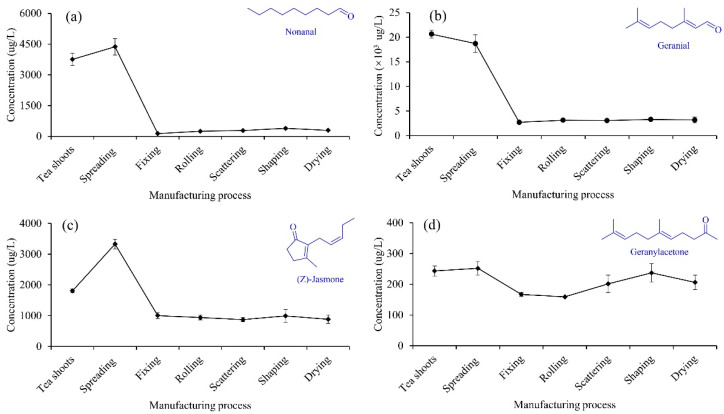
Changes in the concentrations of nonanal (**a**), geranial (**b**), (Z)-jasmone (**c**), and geranylacetone (**d**) during the XYMJ green tea manufacturing.

**Figure 6 foods-11-02682-f006:**
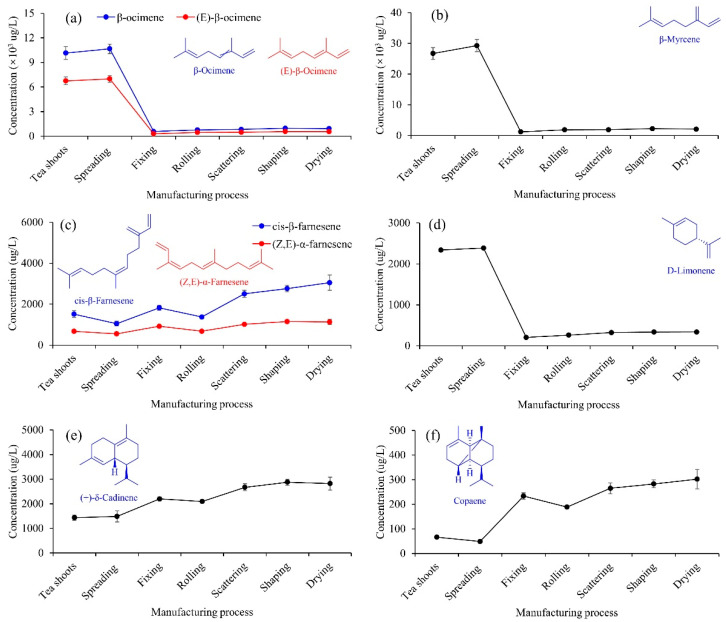
Changes in the concentrations of β-ocimene and (E)-β-ocimene (**a**), β-myrcene (**b**), cis-β-farnesene and (Z,E)-α-farnesene (**c**), D-limonene (**d**), (+)-δ-cadinene (**e**) and copaene (**f**).

**Figure 7 foods-11-02682-f007:**
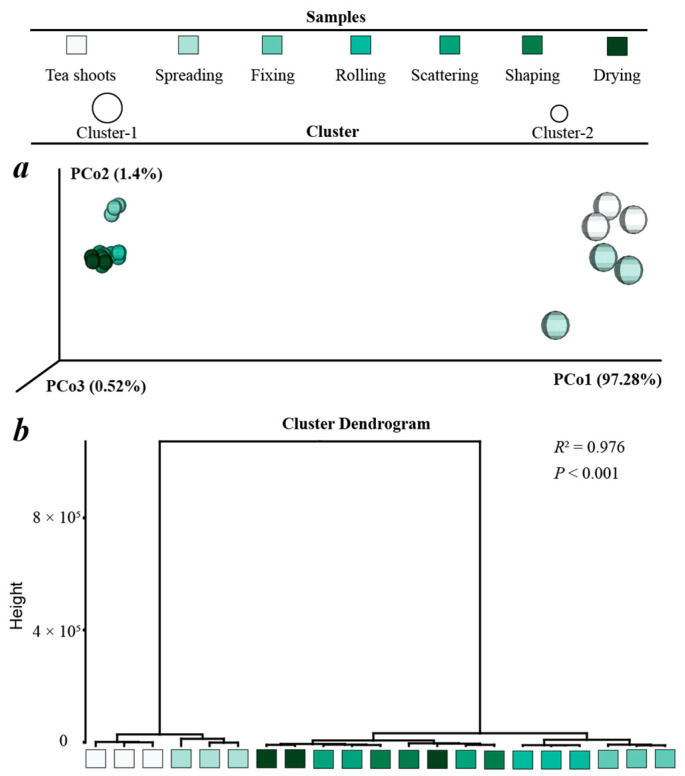
Principal coordinate analysis (**a**) and hierarchical clustering analysis (**b**) of the processed XYMJ green tea samples. The β diversities of volatile compounds were based on Manhattan dissimilarities. (**a**): PCoA scatter plots of the volatile compounds in the processed XYMJ green tea samples, (**b**): the dendrograms represent HCA. PERMANOVA *R*^2^ and *p* values were indicated for the two clusters of the volatile compounds, which were represented by circles of different sizes.

**Figure 8 foods-11-02682-f008:**
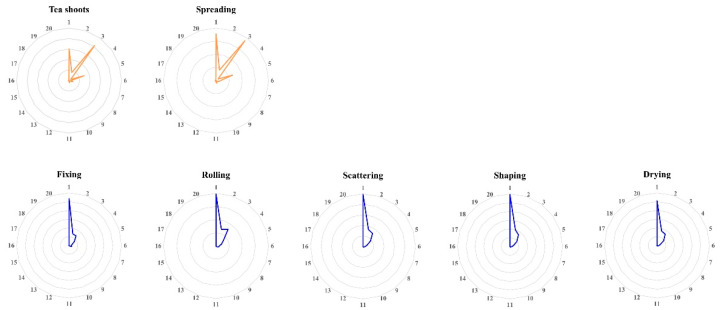
Aroma shape of the processed XYMJ green tea samples based on the top twenty odorants with high OAVs, namely trans-nerolidol (1), geranylacetone (2), nonanal (3), (+)-δ-cadinene (4), linalool (5), (Z)-jasmone (6), cis-3-hexenyl butyrate (7), cis-3-hexenyl hexanoate (8), methyl jasmonate (9), β-ocimene (10), β-myrcene (11), copaene (12), cis-β-farnesene (13), D-limonene (14), (E)-β-ocimene (15), dihydroactinolide (16), M-cymene (17), (Z,E)-α-farnesene (18), methyl salicylate (19), and geranial (20).

**Table 1 foods-11-02682-t001:** Changes in the concentrations of volatile compounds during the XYMJ green tea manufacturing (µg/L).

Compounds	Tea Shoots	Spreading	Fixing	Rolling	Scattering	Shaping	Drying
**Alcohols**							
trans-Furan linalool oxide	5982.4 ± 280.3 ^b^	6638.6 ± 439.8 ^a^	566.3 ± 32.0 ^c^	672 ± 44.4 ^c^	545.2 ± 33.5 ^c^	679.2 ± 70.1 ^c^	596.7 ± 122.9 ^c^
Linalool	45,389.6 ± 1682.8 ^a^	39,633.9 ± 4201.7 ^b^	4265.1 ± 255.0 ^c^	5178.8 ± 303.8 ^c^	4880.6 ± 235.6 ^c^	5453.0 ± 262.1 ^c^	5028.1 ± 876.1 ^c^
(-)-α-Terpineol	853.7 ± 91.5 ^a^	851.5 ± 84.7 ^a^	384.7 ± 37.7 ^c^	370.8 ± 36.0 ^c^	476.7 ± 43.8 ^bc^	562.6 ± 90.2 ^b^	575.9 ± 102.9 ^b^
trans-Nerolidol	3767.5 ± 258.8 ^b^	4422.4 ± 313.7 ^a^	2673.1 ± 172.5 ^c^	1968.8 ± 130.6 ^d^	2477.3 ± 167.3 ^cd^	2959.6 ± 541.4 ^c^	2549.8 ± 447.9 ^cd^
n-Tridecan-1-ol	47.9 ± 2.8 ^a^	48.1 ± 7.5 ^a^	32.6 ± 5.6 ^b^	12.6 ± 2.3 ^c^	8.4 ± 1.9 ^c^	9.3 ± 2.7 ^c^	12.3 ± 2.1 ^c^
cis-Cubenol	158.6 ± 12.8 ^c^	225.2 ± 9.9 ^a^	160.6 ± 3.4 ^c^	151.6 ± 2.1 ^c^	176.0 ± 12.7 ^bc^	196.4 ± 26.7 ^ab^	173.1 ± 19.6 ^bc^
1-Hexadecanol	58.6 ± 8.5 ^a^	44.4 ± 1.7 ^b^	35.2 ± 2.0 ^c^	17.2 ± 1.9 ^d^	9.7 ± 1.4 ^e^	10.6 ± 2.7 ^de^	12.5 ± 1.5 ^de^
2-Ethyldodecan-1-ol	27.3 ± 2.4 ^a^	33.2 ± 4.9 ^a^	21 ± 5.6 ^b^	10.1 ± 0.5 ^c^	9.3 ± 1.0 ^c^	10.7 ± 1.9 ^c^	10.4 ± 1.6 ^c^
Phytol	62.7 ± 16.9 ^e^	116.7 ± 34.6 ^de^	135 ± 9.3 ^de^	187.5 ± 27.7 ^cd^	275.1 ± 64.2 ^bc^	373.4 ± 76.6 ^a^	307.6 ± 56.5 ^ab^
Isophytol	209.0 ± 49.7 ^c^	382.8 ± 161.4 ^bc^	429.5 ± 43.8 ^bc^	615.8 ± 28.9 ^bc^	890.2 ± 199.8 ^ab^	1382.7 ± 305.2 ^a^	950.5 ± 163.0 ^ab^
Dehydroisophytol	28.9 ± 7.1 ^c^	73 ± 21.6 ^bc^	94.9 ± 10.7 ^bc^	120.2 ± 13.0 ^bc^	165.4 ± 69.4 ^ab^	244.5 ± 94.9 ^a^	155.4 ± 53.9 ^ab^
**Esters**							
cis-3-Hexenyl hexanoate	6896.1 ± 374.0 ^a^	4778.9 ± 770.4 ^b^	5328.2 ± 335.3 ^b^	3570.3 ± 76.2 ^c^	3477.6 ± 190.5 ^c^	3518.7 ± 213.8 ^c^	3024.5 ± 558.4 ^c^
(Z)-3-Hexenyl octanoate	1155.7 ± 93.5 ^a^	714.3 ± 87.6 ^b^	712.9 ± 46.3 ^b^	688.0 ± 12.5 ^b^	743.8 ± 62.7 ^b^	794.0 ± 101.9 ^b^	648.4 ± 112.5 ^b^
cis-3-Hexenyl butyrate	1341.9 ± 85.6 ^a^	795.6 ± 117.8 ^b^	466.8 ± 37.2 ^c^	321.9 ± 15.9 ^d^	191.8 ± 10.4 ^e^	179.1 ± 14.8 ^e^	154.5 ± 30.3 ^e^
cis-3-Hexenyl valerate	662.4 ± 41.9 ^a^	249.6 ± 40.3 ^b^	267.4 ± 16.7 ^b^	182.9 ± 10.3 ^c^	178.7 ± 6.6 ^c^	177.7 ± 16.4 ^c^	153.0 ± 33.6 ^c^
Methyl salicylate	11,159.7 ± 1196.8 ^a^	10,675.8 ± 602.8 ^a^	353.1 ± 82.0 ^b^	503.2 ± 53.8 ^b^	278.7 ± 22.4 ^b^	341.2 ± 44.6 ^b^	274.0 ± 124.5 ^b^
Hexadecanoic acid, methyl ester	429.6 ± 28.4 ^a^	276.8 ± 28.0 ^b^	170.7 ± 7.9 ^c^	194.9 ± 10.4 ^c^	201.6 ± 26.4 ^c^	278.7 ± 77.4 ^b^	215.2 ± 26.3 ^bc^
Guaicwood acetate	194.6 ± 15.3 ^c^	206.8 ± 34.4 ^c^	309.8 ± 13.8 ^b^	284.5 ± 5.4 ^b^	397.7 ± 21.1 ^a^	416.0 ± 16.8 ^a^	420.5 ± 36.0 ^a^
(E)-2-Hexenyl hexanoate	666.3 ± 33.4 ^a^	454.7 ± 73.8 ^b^	501.9 ± 37.8 ^b^	417.2 ± 11.1 ^bc^	427.9 ± 21.5 ^b^	430.5 ± 23.1 ^b^	331.1 ± 70.6 ^c^
Hexadecanoic acid, ethyl ester	22.7 ± 2.6 ^a^	23.2 ± 2.1 ^a^	10.4 ± 2.0 ^b^	8.1 ± 0.9 ^b^	10.3 ± 1.6 ^b^	11.0 ± 2.4 ^b^	9.1 ± 1.6 ^b^
cis-3-Hexenyl crotonate	627.6 ± 43.8 ^a^	430.3 ± 52.8 ^b^	320.4 ± 28.7 ^c^	244.8 ± 24.2 ^d^	220.6 ± 24.7 ^d^	228.7 ± 23.6 ^d^	152.7 ± 5.3 ^e^
4-Terpinyl acetate	469.1 ± 28.6 ^a^	478 ± 26.4 ^a^	75 ± 4.8 ^c^	80.4 ± 5.2 ^c^	120.7 ± 5.1 ^b^	121.3 ± 16.8 ^b^	120.5 ± 23.1 ^b^
Hexanoic acid, hexyl ester	231.2 ± 11.6 ^bc^	188.8 ± 31.2 ^cd^	291.5 ± 17.8 ^a^	211.5 ± 3.9 ^bcd^	223.1 ± 12.8 ^bc^	233.3 ± 11.2 ^b^	178.4 ± 38.7 ^d^
Hexadecanoic acid, butyl ester	39.4 ± 18.9 ^bc^	76.3 ± 16.3 ^a^	11.1 ± 6.5 ^c^	26.8 ± 3.3 ^bc^	51.2 ± 18.2 ^ab^	53.9 ± 20.4 ^ab^	48.7 ± 18.0 ^ab^
Isopropyl palmitate	31.1 ± 10.1 ^c^	56.6 ± 15.7 ^ab^	27.7 ± 2.6 ^c^	34.6 ± 1.1 ^bc^	50.0 ± 8.8 ^abc^	58.4 ± 20.7 ^a^	30.8 ± 3.2 ^c^
Dihydroactinolide	64.1 ± 11.3 ^ab^	75.3 ± 7.0 ^a^	31.7 ± 3.5 ^c^	31.3 ± 0.4 ^c^	52.1 ± 2.1 ^abc^	66.5 ± 27.5 ^ab^	48.2 ± 2.4 ^bc^
Methyl jasmonate	673.4 ± 78.7 ^abc^	870.9 ± 87.1 ^a^	367.1 ± 34.4 ^d^	393.8 ± 71.1 ^cd^	456.9 ± 105.5 ^bcd^	690.6 ± 79.6 ^ab^	375.3 ± 45.0 ^d^
Phytol acetate	667.1 ± 220.5 ^d^	1345.7 ± 422.2 ^cd^	1506.6 ± 88.2 ^cd^	2118.5 ± 363.5 ^bc^	2983.2 ± 428.8 ^b^	4279.8 ± 510.7 ^a^	3293.3 ± 578.1 ^ab^
**Terpenes**							
β-Ocimene	10,145.2 ± 777.1 ^b^	10,668.8 ± 584.5 a	573.1 ± 40.6 ^c^	760.0 ± 44.7 ^c^	826.6 ± 63.9 ^c^	965.0 ± 52.2 ^c^	924.1 ± 72.6 ^c^
β-Myrcene	26,715.1 ± 1929.3 ^b^	29,265.3 ± 1984.2 a	1208.8 ± 68.3 ^c^	1879 ± 161.6 ^c^	1891.8 ± 137.4 ^c^	2235.6 ± 184.1 ^c^	2094.8 ± 92.8 ^c^
D-Limonene	2342.0 ± 246.5 ^b^	2388.8 ± 166.6 a	206.2 ± 10.7 ^c^	261.0 ± 14.9 ^c^	324.4 ± 18.3 ^c^	336.3 ± 27.1 ^c^	338.3 ± 28.8 ^c^
cis-β-Farnesene	1512.5 ± 167.1 ^cd^	1047.1 ± 113.0 e	1819.1 ± 115.8 ^cd^	1373 ± 16.4 ^e^	2505.2 ± 172.3 ^b^	2759.2 ± 165.4 ^ab^	3047.7 ± 371.7 ^a^
(+)-δ-Cadinene	1435.8 ± 104.3 ^c^	1485.9 ± 227.7 ^c^	2199.0 ± 72.0 ^b^	2095.6 ± 30.2 ^b^	2666.4 ± 142.4 ^a^	2875.9 ± 126.3 ^a^	2821.5 ± 261.7 ^a^
α-Cubebene	215.1 ± 20.8 ^c^	293.8 ± 15.0 ^ab^	258.0 ± 7.8 ^b^	257.6 ± 12.5 ^b^	291.5 ± 26.9 ^ab^	294.7 ± 14.8 ^ab^	303.1 ± 34.1 ^a^
(E)-β-Ocimene	6744.4 ± 464.7 ^a^	6993.1 ± 399.2 ^a^	299.4 ± 19.5 ^b^	455.9 ± 26.5 ^b^	478.8 ± 34.6 ^b^	565.8 ± 35.8 ^b^	552.0 ± 13.4 ^b^
Cubenene	384.3 ± 31.7 ^d^	462.5 ± 46.5 ^cd^	588.0 ± 18.4 ^b^	534.6 ± 8.7 ^bc^	689.7 ± 36.8 ^a^	759.5 ± 52.4 ^a^	747.5 ± 58.3 ^a^
(-)-α-Bergamotene	267.0 ± 32.9 ^c^	204.5 ± 25.1 ^c^	383.8 ± 17.8 ^b^	272 ± 7.5 ^c^	502.3 ± 34.6 ^a^	564.7 ± 49.9 ^a^	581.2 ± 82.8 ^a^
α-Calacorene	370.5 ± 31.6 ^d^	417.1 ± 33.8 ^d^	483.9 ± 10.5 ^c^	478.4 ± 13.8 ^c^	550.8 ± 29.4 ^b^	630.6 ± 25.4 ^a^	621.5 ± 32.4 ^a^
cis-Calamenene	985.8 ± 73.3 ^c^	1093 ± 113.2 ^c^	1688.3 ± 51.3 ^b^	1676.4 ± 45.1 ^b^	2025.8 ± 128 ^a^	2041.3 ± 119.4 ^a^	2103 ± 185.2 ^a^
(+)-4-Carene	616.2 ± 49.3 ^a^	590.6 ± 47.0 ^a^	54.4 ± 0.9 ^b^	66.3 ± 4.1 ^b^	82.3 ± 4.7 ^b^	85.1 ± 8.6 ^b^	84.2 ± 11.0 ^b^
Cadina-4(14),5-diene	315.3 ± 18.1 ^c^	356 ± 52.1 ^c^	375.8 ± 23.9 ^bc^	381.7 ± 18.7 ^bc^	459.6 ± 35.7 ^ab^	510.5 ± 45.7 ^a^	443.3 ± 72.0 ^ab^
(Z,E)-α-Farnesene	672.0 ± 76.3 ^c^	551.3 ± 82.8 ^c^	920.9 ± 45.1 ^b^	677.4 ± 14.6 ^c^	1014.1 ± 69.4 ^ab^	1150.1 ± 82.5 ^a^	1129.0 ± 126.0 ^a^
Copaene	66.3 ± 3.5 ^d^	48.8 ± 3.1 ^d^	233.4 ± 12.9 ^b^	188.8 ± 2.1 ^c^	265.0 ± 22.1 ^ab^	282.8 ± 16.3 ^a^	302.4 ± 39.4 ^a^
(E)-α-Bisabolene	61.1 ± 10.3 ^a^	55 ± 6.4 ^ab^	43.6 ± 3 ^cd^	35.9 ± 2.5 ^d^	47.9 ± 4.6 ^bc^	52.9 ± 4.0 ^abc^	57.3 ± 2.0 ^ab^
β-Calacorene	58.0 ± 5.8 ^e^	62.7 ± 9.7 ^de^	73.6 ± 2.9 ^bcd^	72.2 ± 2.5 ^cd^	81.5 ± 5.8 ^abc^	93.6 ± 8.5 ^a^	85.6 ± 7.8 ^ab^
α-Thujene	816.9 ± 77.9 ^a^	781 ± 33.2 ^a^	67.7 ± 3.2 ^b^	87.6 ± 9.3 ^b^	84.2 ± 4.2 ^b^	98.5 ± 3.9 ^b^	106.5 ± 15.3 ^b^
β-Bisabolene	73.8 ± 11.5 ^abc^	70.5 ± 5.0 ^bc^	63.3 ± 5.0 ^cd^	51.5 ± 2.2 ^d^	72.3 ± 8.7 ^abc^	78.8 ± 2.7 ^ab^	84.2 ± 1.5 ^a^
**Ketones**							
(Z)-Jasmone	1806.1 ± 63.4 ^b^	3330.3 ± 156.7 ^a^	1004.2 ± 100.9 ^c^	938.8 ± 76.9 ^c^	871.6 ± 66.6 ^c^	994.1 ± 205.1 ^c^	884.7 ± 143.9 ^c^
Phytone	192.1 ± 27.3 ^ab^	212 ± 17.2 ^ab^	151.7 ± 13.1 ^b^	165.8 ± 18.4 ^b^	204.4 ± 32.1 ^ab^	265.3 ± 86.6 ^a^	201.4 ± 22.3 ^ab^
Geranylacetone	243.4 ± 16.4 ^a^	251.8 ± 22.4 ^a^	166.9 ± 5.8 ^b^	159.2 ± 2.5 ^b^	201.3 ± 28.6 ^ab^	237.0 ± 29.8 ^a^	206.2 ± 23.4 ^ab^
**Aldehydes**							
Geranial	3756.4 ± 292.6 ^b^	4372.1 ± 398.3 ^a^	134.3 ± 11.8 ^c^	246.9 ± 30.2 ^c^	281.7 ± 29.7 ^c^	390.2 ± 51.8 ^c^	294.1 ± 3.9 ^c^
Nonanal	20,634.3 ± 793.8 ^a^	18,717.2 ± 1798.9 ^b^	2701.0 ± 125.7 ^c^	3132.8 ± 135.6 ^c^	3065.1 ± 153.8 ^c^	3292.8 ± 138.7 ^c^	3190.1 ± 591.9 ^c^
**Hydrocarbons**							
Hexadecane	566.4 ± 85.5 ^a^	540.9 ± 97.6 ^a^	319.5 ± 64.7 ^b^	174.7 ± 7.3 ^c^	121.8 ± 13.1 ^c^	162.4 ± 31.4 ^c^	201.1 ± 23.5 ^c^
3,4-Dimethyl-2,4,6-octatriene	1658.3 ± 139.4 ^a^	1684.3 ± 112.3 ^a^	74.9 ± 5.3 ^b^	1110.0 ± 8.2 ^b^	120 ± 9.9 ^b^	139.7 ± 11.6 ^b^	128.1 ± 4.9 ^b^
Tetradecane	326.8 ± 24.6 ^bc^	364.0 ± 9.1 ^ab^	337.5 ± 17.3 ^bc^	220.9 ± 21.7 ^e^	244.2 ± 23.4 ^de^	286.0 ± 10.1 ^cd^	409.1 ± 51.2 ^a^
Dodecane	211.6 ± 30.1 ^ab^	207.1 ± 28.6 ^ab^	203 ± 21.1 ^ab^	139.9 ± 30.9 ^c^	167.2 ± 6.2 ^bc^	214.3 ± 28.0 ^ab^	223.5 ± 33.8 ^a^
Pentadecane	180.3 ± 27.5 ^ab^	232.3 ± 51.9 ^a^	189.8 ± 45.6 ^ab^	145.1 ± 12.7 ^b^	162.6 ± 3.4 ^b^	194.4 ± 13.9 ^ab^	169.1 ± 16.9 ^b^
Dodecane, 6-methyl-	75.0 ± 7.5 ^a^	60.1 ± 18.1 ^a^	74.5 ± 12.1 ^a^	59.1 ± 13.8 ^a^	58.7 ± 3.1 ^a^	70.8 ± 10.7 ^a^	60.7 ± 11.0 ^a^
2,6,10-Trimethyltridecane	848.1 ± 80.2 ^a^	654.5 ± 145 ^b^	362.7 ± 20.9 ^c^	311.8 ± 29.9 ^c^	386.6 ± 34.4 ^c^	446.1 ± 19.3 ^c^	414.4 ± 47.8 ^c^
Eicosane	322.6 ± 67.9 ^abc^	354.9 ± 100.1 ^ab^	212.7 ± 27.1 ^bc^	184.6 ± 20.0 ^c^	346.3 ± 109.7 ^ab^	401.3 ± 116.4 ^a^	175.7 ± 28.4 ^c^
Dodecane, 4,6-dimethyl-	100.9 ± 19.2 ^a^	133.8 ± 29.8 ^a^	116.0 ± 19.3 ^a^	112.1 ± 20.0 ^a^	115.0 ± 9.6 ^a^	133.7 ± 11.1 ^a^	107.1 ± 16.3 ^a^
Undecane, 4,7-dimethyl-	109.6 ± 17.6 ^a^	129.5 ± 21.1 ^a^	115.8 ± 19.2 ^a^	105.4 ± 21.9 ^a^	110.9 ± 13.0 ^a^	130.1 ± 11.5 ^a^	107.6 ± 14.3 ^a^
Heptadecane	220.1 ± 34.2 ^b^	311.1 ± 80.3 ^a^	148.8 ± 20.4 ^c^	109.4 ± 18.6 ^c^	113.7 ± 7.9 ^c^	148.0 ± 19.0 ^c^	104.1 ± 5.2 ^c^
(E)-4,8-Dimethyl-1,3,7-nonatriene	103.1 ± 14.1 ^c^	156.3 ± 12.2 ^c^	1074.8 ± 92.9 ^a^	581.9 ± 39.6 ^b^	1151.7 ± 7.5 ^a^	1161.7 ± 209.1 ^a^	1179.9 ± 300.5 ^a^
Undecane, 3,8-dimethyl-	62.0 ± 7.1 ^a^	45.9 ± 5.9 ^a^	54.6 ± 7.8 ^a^	49.4 ± 9.5 ^a^	51.6 ± 6.6 ^a^	55.7 ± 6.2 ^a^	57.9 ± 15.1 ^a^
Undecane, 2,8-dimethyl	26.5 ± 1.0 ^a^	20.0 ± 7.2 ^a^	25.1 ± 4.1 ^a^	29.1 ± 0.7 ^a^	22.1 ± 1.6 ^a^	26.5 ± 3.5 ^a^	25.4 ± 6.7 ^a^
Nonane, 4,5-dimethyl-	76.0 ± 11.5 ^a^	76.7 ± 10.5 ^a^	67.4 ± 9.7 ^a^	64.9 ± 12.7 ^a^	67.9 ± 3.8 ^a^	76.7 ± 10.5 ^a^	61 ± 10.4 ^a^
4,6-Dimethylundecane	24.9 ± 0.9 ^d^	25.5 ± 7.5 ^cd^	37.9 ± 6.2 ^abc^	41.5 ± 2.3 ^ab^	43.4 ± 4.4 ^ab^	45.2 ± 11.5 ^a^	30.9 ± 4.8 ^bcd^
Undecane, 3,6-dimethyl-	80.3 ± 6.4 ^a^	76.4 ± 15.1 ^ab^	70.4 ± 12.0 ^ab^	57.5 ± 9.1 ^b^	63.9 ± 9.1 ^ab^	75.4 ± 5.9 ^ab^	68.1 ± 8.0 ^ab^
Decane, 3-ethyl-3-methyl-	38.1 ± 4.1 ^a^	37.1 ± 9.3 ^a^	28.2 ± 3.9 ^ab^	19.0 ± 3.4 ^b^	28.3 ± 4.6 ^ab^	31.7 ± 1.3 ^a^	35.0 ± 6.4 ^a^
Undecane, 3-ethyl-	37.5 ± 0.1 ^a^	36.0 ± 7.1 ^a^	36.5 ± 3.7 ^a^	28.1 ± 4.6 ^a^	30.1 ± 4.4 ^a^	36.6 ± 1.5 ^a^	35.4 ± 5.7 ^a^
M-Cymene	781.2 ± 44.4 ^a^	787.6 ± 64.8 ^a^	121 ± 25.1 ^b^	124.8 ± 13.5 ^b^	153.0 ± 0.7 ^b^	146.6 ± 12.5 ^b^	150.4 ± 12.3 ^b^
1,3-Diisopropylbenzene	21.5 ± 0.8 ^d^	36.1 ± 6.9 ^c^	39.9 ± 1.6 ^bc^	42.3 ± 4.6 ^bc^	47.4 ± 2.1 ^ab^	55.9 ± 4.2 ^a^	53.2 ± 9.8 ^a^

Note: All data are expressed as mean ± S.D. (*n* = 3). Different letters in the same row indicate significant differences between mean values (*p* < 0.05, ANOVA, Duncan test).

## Data Availability

Not applicable.

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
