# Peer review of "Dynamic Changes of Volatile Compounds during the Xinyang Maojian Green Tea Manufacturing at an Industrial Scale"

_foods, 2022, doi:10.3390/foods11172682_

Round 1
Reviewer 1 Report
1) Experimental material (lines 89-108), the authors are kindly asked to cite the reference they used for the steps. If the process is novel, this must be mentioned in addition to citing others who followed similar procedure.
2) The authors are kindly asked to mention the reference they were based on for the brewing time and temperature (lines 113-124). Please mention if you tried any other conditions like different temperatures or brewing time.
3) The authors are kindly asked to cite the reference when discussing the method (lines 126-148).
4) Did the authors use any standards to identify the mentioned compounds or the identification was based on the library of the GC-MS? Kindly please mention this.
5) Please mention the reason of the change of the concentration (line 267).
Author Response
Dear Editor and Reviewers, please see the attachment.

Reviewer 2 Report
1-At the beginning of the summary, what was done suddenly was explained, and without saying we did it in this study per sentence, it came down from the top. (Line 18-19)
2-More comprehensive information about the tea mentioned in the introduction to the abstract.
https://en.wikipedia.org/wiki/Xinyang_Maojian_tea
3-I would like to see what this tea looks like alone.
4- The sentence given at the end of the introduction could also be given in the summary. (Line 82-83)
‘Thus, the aim of the present study was to investigate the key odorants in XYMJ green tea and their dynamic changes during the manufacturing processes using HS-SPME combined with GC-MS.’
5-Tables are vertical. It could be horizontal.
6- No mention of HS-SPME. What it is and how to do it is not explained.
7- Studies in the literature are lacking. It seems like the information about who did what and what he found is missing.
Author Response

(The authors gave the same response as above.)
